# Integrating Computational Design and Experimental Approaches for Next-Generation Biologics

**DOI:** 10.3390/biom14091073

**Published:** 2024-08-27

**Authors:** Ahrum Son, Jongham Park, Woojin Kim, Wonseok Lee, Yoonki Yoon, Jaeho Ji, Hyunsoo Kim

**Affiliations:** 1Department of Molecular Medicine, Scripps Research, La Jolla, CA 92037, USA; ahson@scripps.edu; 2Department of Bio-AI Convergence, Chungnam National University, 99 Daehak-ro, Yuseong-gu, Daejeon 34134, Republic of Korea; 975pjh@gmail.com (J.P.); woojin1544@gmail.com (W.K.); wonseogi46@gmail.com (W.L.); dbsrl0218@gmail.com (Y.Y.); 3Department of Convergent Bioscience and Informatics, Chungnam National University, 99 Daehak-ro, Yuseong-gu, Daejeon 34134, Republic of Korea; kkdjaeho0612@gmail.com; 4Protein AI Design Institute, Chungnam National University, 99 Daehak-ro, Yuseong-gu, Daejeon 34134, Republic of Korea; 5SCICS (Sciences for Panomics), 99 Daehak-ro, Yuseong-gu, Daejeon 34134, Republic of Korea

**Keywords:** protein design, protein engineering, protein therapeutics, antibody engineering, cytokine engineering, enzyme replacement therapy

## Abstract

Therapeutic protein engineering has revolutionized medicine by enabling the development of highly specific and potent treatments for a wide range of diseases. This review examines recent advances in computational and experimental approaches for engineering improved protein therapeutics. Key areas of focus include antibody engineering, enzyme replacement therapies, and cytokine-based drugs. Computational methods like structure-based design, machine learning integration, and protein language models have dramatically enhanced our ability to predict protein properties and guide engineering efforts. Experimental techniques such as directed evolution and rational design approaches continue to evolve, with high-throughput methods accelerating the discovery process. Applications of these methods have led to breakthroughs in affinity maturation, bispecific antibodies, enzyme stability enhancement, and the development of conditionally active cytokines. Emerging approaches like intracellular protein delivery, stimulus-responsive proteins, and de novo designed therapeutic proteins offer exciting new possibilities. However, challenges remain in predicting in vivo behavior, scalable manufacturing, immunogenicity mitigation, and targeted delivery. Addressing these challenges will require continued integration of computational and experimental methods, as well as a deeper understanding of protein behavior in complex physiological environments. As the field advances, we can anticipate increasingly sophisticated and effective protein therapeutics for treating human diseases.

## 1. Introduction

Protein-based therapeutics have revolutionized medicine over the past few decades, offering highly specific and potent treatments for a wide range of diseases [1,2]. As of 2023, over 350 protein-based drugs have been approved for clinical use, with many more in development [3]. The success of protein therapeutics can be attributed to their ability to perform complex biological functions with high specificity and low toxicity compared to small-molecule drugs [2,3]. However, natural proteins often lack optimal pharmaceutical properties such as stability, half-life, and manufacturability [1,4]. Protein design and engineering approaches have emerged as powerful tools to overcome these limitations and create improved biotherapeutics with enhanced efficacy, safety, and developability [5].

The field of therapeutic protein engineering has expanded rapidly, driven by advances in computational modeling, high-throughput screening techniques, and our deepening understanding of protein structure–function relationships [3,5,6]. These developments have enabled researchers to modify existing proteins and even create entirely novel protein structures tailored for specific therapeutic applications. Key areas of focus include antibody engineering, enzyme replacement therapies, and the development of cytokine-based drugs [3,5].

Computational protein design has played an increasingly important role in this field [7]. Tools such as Rosetta, RoseTTAFold, and RF Diffusion have dramatically improved our ability to predict protein structures, design stable proteins, and engineer proteins for specific molecular interactions [8]. These computational approaches, when combined with experimental validation, have led to breakthroughs such as the de novo design of protein binders, enzymes with novel catalytic activities, and protein-based vaccines [3].

Experimental protein engineering techniques have also seen significant advancements [9,10]. Directed evolution methods, including phage display and yeast surface display, have been refined to rapidly evolve proteins with desired properties [3,11,12]. Additionally, the integration of non-canonical amino acids and chemical modifications has expanded the toolkit available for protein engineering, enabling the creation of biotherapeutics with enhanced stability, pharmacokinetics, and novel functionalities [11,13,14,15,16].

Building upon these advancements, the development of bispecific and multi-specific antibodies exemplifies the potential of therapeutic protein engineering to create next-generation biologics capable of addressing complex disease mechanisms [17]. These engineered proteins can simultaneously bind to multiple targets, offering innovative solutions in areas such as cancer immunotherapy and other multifactorial diseases [18]. Other emerging applications include the design of intracellular protein therapeutics, conditionally activated proteins, and protein-based nanocarriers for drug delivery [19,20,21].

The fusion of computational and experimental techniques is essential in the field of therapeutic protein engineering [3]. Ultra-high-throughput screening is a cost-effective and impartial method to select interesting candidates for further engineering. By combining experimental methods with structural investigations, computational methodologies can be enhanced to more precisely forecast protein behavior and function. The combination of computational design and experimental validation not only improves the accuracy of protein engineering but also speeds up the creation of new therapies. To overcome the current obstacles in computational design accuracy, experimental screening efficiency, immunogenicity, and manufacturing scalability, it is crucial to maintain ongoing collaboration within these fields [22]. This collaboration is necessary to make significant advancements in targeted delivery techniques and therapeutic effectiveness [23].

This review will examine recent advances in computational and experimental protein engineering methods and their applications in developing next-generation protein therapeutics. We will discuss strategies for optimizing protein stability, pharmacokinetics, targeting, and functionality, as well as emerging approaches for creating novel protein-based drugs with unique capabilities. For each application, we will explore the specific challenges addressed by protein engineering and highlight notable successes and ongoing clinical trials. We will also discuss the integration of protein engineering with other emerging technologies, such as cell and gene therapies. Finally, we will explore future directions and challenges in the field of therapeutic protein engineering, including emerging computational tools, novel experimental techniques, and potential new therapeutic modalities. We will also consider the broader implications of advances in protein engineering for personalized medicine and the development of treatments for currently intractable diseases.

By providing a comprehensive overview of the current state of the art in therapeutic protein engineering, this review aims to serve as a valuable resource for researchers, clinicians, and biotechnology professionals working at the forefront of this rapidly evolving field.

## 2. Computational Protein Design

### 2.1. Structure-Based Design

Structure-based computational design has become an invaluable tool for engineering therapeutic proteins with improved properties [7]. This approach leverages available protein structural data and physics-based modeling to predict the effects of amino acid mutations on protein stability, binding affinity, and function [24] (Figure 1a).

#### 2.1.1. Machine Learning Integration

The integration of machine learning, particularly deep learning models, has revolutionized computational protein engineering by dramatically improving protein structure prediction and design capabilities [25,26]. AlphaFold, developed by DeepMind, has achieved unprecedented accuracy in predicting protein structures from amino acid sequences, with many predictions reaching atomic-level precision. This breakthrough has accelerated research across structural biology and enabled new approaches to protein design and engineering. The success of AlphaFold has inspired the development of other AI-powered tools for protein structure prediction and design, such as RoseTTAFold and ESMFold, further expanding the toolkit available to researchers [27,28]. Integration of these deep learning models with traditional physics-based algorithms is enhancing both the accuracy and scope of computational protein engineering [29]. For example, researchers have developed methods to incorporate physics-based force fields as differentiable modules within deep learning frameworks, allowing for more physically realistic predictions and designs. This synergy between data-driven machine learning approaches and physics-based modeling is enabling more robust and reliable computational protein engineering pipelines. The impact of these advancements extends beyond structure prediction to areas such as protein–protein interaction prediction, enzyme design, and drug discovery, opening up new possibilities for creating novel proteins with tailored functions [26]. As the field continues to evolve, the integration of machine learning with experimental techniques and high-throughput screening methods promises to further accelerate the discovery and optimization of engineered proteins for therapeutic and biotechnological applications.

#### 2.1.2. AlphaFold vs. RosettaFold

AlphaFold and Rosetta are two leading tools in the field of protein structure prediction, each with distinct methodologies and applications. AlphaFold, developed by DeepMind, employs a deep learning approach that leverages sequence coevolution data to predict the distance and angle between amino acids, achieving remarkable accuracy in protein folding predictions. This method has demonstrated exceptional performance in the Critical Assessment of Structure Prediction (CASP) competitions, with AlphaFold2 achieving a median Global Distance Test (GDT) score of 92.4, close to experimental accuracy [30]. In contrast, Rosetta, created by the Baker lab, uses a combination of physics-based and knowledge-based methods, employing Monte Carlo algorithms to sample protein conformations and score them based on their probability. Rosetta is known for its flexibility and has been extensively used for protein design, docking, and related tasks [31]. While AlphaFold excels in predicting monomeric protein structures with high accuracy, Rosetta offers robust performance in modeling protein complexes and quaternary structures, especially when supplemented with experimental data such as covalent labeling. However, AlphaFold’s reliance on neural networks can sometimes lead to inaccuracies in loop regions and dynamic binding sites, where Rosetta’s physics-based approach may provide more reliable results [30,32]. Despite AlphaFold’s groundbreaking accuracy, it occasionally struggles with predicting the structural impact of point mutations and conformational ensembles, areas where Rosetta’s energy-based scoring and conformational sampling can offer more detailed insights [33]. Overall, while AlphaFold represents a significant advancement in AI-driven protein structure prediction, Rosetta’s comprehensive toolkit and integration with experimental data make it a valuable complement, particularly for complex and dynamic protein systems.

#### 2.1.3. Rosetta Software Suite

The Rosetta software (version 3.14) suite is a comprehensive platform for macromolecular modeling, docking, and design that has been extensively developed over two decades by a global community of researchers [34]. It includes algorithms for computational modeling and analysis of protein structures, enabling notable scientific advances in areas such as de novo protein design, enzyme design, ligand docking, and structure prediction of biological macromolecules and complexes. Originally developed in the laboratory of David Baker at the University of Washington for protein structure prediction, Rosetta has since expanded to address a wide range of computational challenges in structural biology. Recent applications of Rosetta include the design of miniprotein binders against targets like SARS-CoV-2 and influenza hemagglutinin [35]. The software has been continuously refined and extended, with recent developments incorporating deep learning techniques such as RoseTTAFold for rapid protein structure prediction and RFdiffusion for generative protein design [8]. Rosetta is freely available to academic and non-profit users, while commercial entities can obtain licenses through the University of Washington [34]. The collaborative nature of Rosetta’s development, involving researchers from over 60 institutions, has contributed to its widespread use and ongoing innovation in the field of computational structural biology.

### 2.2. Sequence-Based Design

Complementing structure-based methods, sequence-based computational approaches leverage the wealth of genomic and protein sequence data to guide protein engineering efforts (Figure 1b).

#### 2.2.1. Machine Learning on Sequence Data

Deep learning models trained on large protein sequence databases have emerged as powerful tools for predicting the effects of mutations and guiding directed evolution experiments in protein engineering. Convolutional neural networks built with amino acid property descriptors have demonstrated strong performance in predicting protein redesign outcomes across diverse datasets [36]. These models can efficiently screen large numbers of novel sequences in silico, accelerating the protein engineering process. Notable examples of generative models for protein sequences include Protein-GAN, which uses generative adversarial networks to expand functional protein sequence spaces, and ProteinMPNN, a graph neural network approach for designing stable and functional de novo proteins [8]. ProteinMPNN in particular has shown outstanding performance in both computational and experimental tests, with higher native sequence recovery (52.4%) compared to traditional methods like Rosetta (32.9%) when redesigning protein backbones [37]. It can generate sequences for complex protein structures including monomers, cyclic homo-oligomers, and binding proteins [38]. The model incorporates noise during training to improve robustness when designing sequences for predicted protein structures [39]. ProteinMPNN has been successfully applied to rescue previously failed designs of various protein architectures, demonstrating its broad utility [40]. By combining sequence-based machine learning with structure-based methods and experimental validation, researchers are developing increasingly accurate computational pipelines for engineering proteins with enhanced stability, binding affinity, and catalytic activity [41].

#### 2.2.2. Language Models for Proteins

Large language models trained on protein sequences have emerged as powerful tools for various protein engineering tasks. ESM-1b, developed by Meta AI (formerly Facebook AI Research), is a prominent example of such models, trained on 250 million protein sequences using masked language modeling [42]. This model has demonstrated impressive capabilities in predicting protein properties, and functions directly from individual sequences ESM-1b can be used for tasks such as predicting the effects of mutations, inferring protein structure, and annotating protein function [42]. The model’s learned representations have been shown to contain information about protein secondary and tertiary structure, which can be extracted through linear projections. ESM-1b has achieved state-of-the-art performance in zero-shot prediction of mutational effects and secondary structure prediction. The model’s success has led to the development of more advanced versions, such as ESM-2 and ESMFold, which enable even more accurate structure prediction. These language models have also been applied to tasks like protein design, with tools like ESM-IF1 for inverse folding. Recent work has explored fine-tuning these models on small amounts of experimental data to enhance their predictive capabilities for specific proteins or properties [43]. The integration of these language models with other computational and experimental techniques is advancing the field of protein engineering, enabling more efficient exploration of protein sequence space and the design of proteins with desired properties [44].

#### 2.2.3. Comparative Analysis of Sequence-Based vs. Structure-Based Drug Design

Recent advancements in drug discovery have highlighted the distinct methodologies and applications of sequence-based and structure-based design approaches, each with its unique strengths and limitations. Sequence-based drug design, such as the sequence-to-drug concept validated by TransformerCPI2.0, leverages protein sequence data to predict drug candidates through end-to-end differentiable learning models. This method is particularly advantageous for targets lacking high-resolution 3D structures, as demonstrated by its success in identifying novel modulators for proteins like SPOP and RNF130, which are challenging to model structurally [45]. On the other hand, structure-based drug design (SBDD) relies on the detailed 3D structures of target proteins to identify binding sites and optimize drug interactions. Recent advances in computational algorithms and techniques like cryo-electron microscopy have significantly enhanced the precision of SBDD, enabling the design of highly specific and effective biologics [46,47]. However, SBDD is often limited by the availability and accuracy of protein structures and the complexity of modeling dynamic binding sites [47,48]. While sequence-based methods offer a more direct and potentially faster route to drug discovery, structure-based approaches provide deeper insights into molecular interactions, crucial for optimizing drug efficacy and safety. The integration of both methods, leveraging the strengths of each, holds promise for advancing the development of next-generation therapeutics, offering a comprehensive toolkit for tackling diverse drug discovery challenges [45,46].

## 3. Experimental Protein Engineering

### 3.1. Directed Evolution

Directed evolution remains a cornerstone of protein engineering, allowing the creation of proteins with dramatically enhanced or novel functions through iterative rounds of mutation and selection (Figure 2a).

#### 3.1.1. Phage-Assisted Continuous Evolution

Phage-assisted continuous evolution (PACE) is a powerful system that enables rapid directed evolution of proteins and other biomolecules. PACE exploits the life cycle of M13 bacteriophage to continuously evolve gene-encoded molecules that can be linked to phage production in *E. coli* host cells [49,50]. The system allows for continuous mutagenesis, selection, and replication without researcher intervention, enabling hundreds of rounds of evolution to occur in days or weeks rather than months. PACE achieves this acceleration by coupling the desired activity to the production of phage infectivity protein pIII, which is essential for phage propagation. Mutagenesis is driven by an error-prone DNA polymerase expressed in the host cells, generating diversity continuously during phage replication [51]. The use of a “lagoon” vessel with continuous influx of fresh host cells and outflow of depleted cells allows evolution to proceed indefinitely [52]. PACE has been successfully applied to evolve a wide range of proteins, including polymerases, proteases, genome editing tools, and antibody fragments, often yielding variants with dramatically improved properties after just days of evolution [49]. The system can be modulated through strategies like negative selection or modifying export levels to the periplasm, allowing fine-tuning of selection stringency. Overall, PACE represents a significant advance in directed evolution technology, enabling the rapid generation of biomolecules with novel and enhanced functions (Figure 2b).

#### 3.1.2. Deep Mutational Scanning

Deep mutational scanning (DMS) is a high-throughput approach that systematically assesses the effects of all possible single amino acid substitutions on protein function, providing comprehensive mutational landscapes to guide engineering efforts [53,54,55]. DMS involves creating large libraries of protein variants containing thousands to millions of mutations, followed by selection or screening and high-throughput sequencing to quantify the functional effects of each mutation [53,54]. This technique enables the exploration of protein sequence–function relationships at an unprecedented scale and can reveal intrinsic protein properties, protein behavior within cells, and the consequences of genetic variations. DMS has been applied to study protein stability, binding affinity, enzymatic activity, and other functional properties across diverse proteins including enzymes, antibodies, and viral proteins [56]. The comprehensive mutational data generated by DMS can guide rational protein engineering by identifying key functional residues, revealing permissive sites for modification, and uncovering non-obvious beneficial mutations. Additionally, DMS data can be used to train machine learning models for predicting the effects of mutations, further enhancing protein engineering capabilities [53,54]. Recent advances in DMS methodology, including the use of CRISPR-based genome editing for in situ mutagenesis and the development of more sophisticated selection schemes, have expanded the applicability of this approach to studying proteins in their native genomic context [56] (Figure 2b).

### 3.2. Rational Design and Structure-Guided Engineering

Rational design approaches leverage structural and mechanistic insights to make targeted modifications to proteins (Figure 2a).

#### 3.2.1. Computational–Experimental Hybrid Approaches

Computational–experimental hybrid approaches, involving iterative cycles of computational prediction and experimental validation, have emerged as powerful strategies for efficiently optimizing protein properties. These approaches leverage the strengths of both computational modeling and experimental testing to accelerate the protein engineering process [57,58]. Typically, the workflow begins with computational predictions of promising protein variants, which are then experimentally tested [59]. The experimental results are used to refine the computational models, creating an iterative feedback loop that improves prediction accuracy over multiple rounds [57]. This approach has been successfully applied to various protein engineering challenges, including enhancing enzyme activity, stability, and specificity. For example, Voigt et al. demonstrated the power of this method by rapidly optimizing β-lactamase activity using a combination of structure-guided computational design and high-throughput screening [60]. The integration of machine learning techniques with experimental data has further enhanced the efficiency of these hybrid approaches, enabling more accurate predictions of protein properties from limited experimental data [61]. Additionally, these methods can incorporate diverse types of experimental data, including structural information from X-ray crystallography, NMR, and cryo-EM, as well as functional data from high-throughput assays [62]. By combining computational and experimental techniques, researchers can explore larger sequence spaces more efficiently than through experimental methods alone, while also overcoming limitations of purely computational approaches [58] (Figure 2c).

#### 3.2.2. Protein Resurfacing

Protein resurfacing is a powerful protein engineering approach that involves modifying surface residues to improve stability, reduce immunogenicity, and alter pharmacokinetics without disrupting the core protein function [63,64]. This technique allows for extensive modification of protein surfaces, with some studies achieving up to 58 substitutions, resulting in direct modification of 35% of surface residues [19]. Resurfacing can significantly reduce binding to pre-existing anti-drug antibodies, with greater reductions observed as mutational distance from the native protein increases [15]. In addition to reducing immunogenicity, resurfacing can enhance protein stability by optimizing surface charge distribution and introducing favorable electrostatic interactions [65,66]. The approach has been successfully applied to various therapeutic proteins, including enzymes like L-asparaginase, to generate variants with improved properties while maintaining catalytic function [1]. Computational design methods play a crucial role in resurfacing efforts, allowing researchers to explore large sequence spaces and predict mutations that maintain protein folding and function [67,68]. When combined with experimental validation, these computational approaches enable the efficient development of resurfaced proteins with enhanced pharmaceutical properties. As the field advances, machine learning techniques and large-scale design–test–learn cycles are likely to further improve our ability to optimize the trade-offs between immunogenicity, function, and expression in resurfaced proteins [19,41] (Figure 2c).

## 4. Applications in Therapeutic Protein Engineering

### 4.1. Antibody Engineering

Monoclonal antibodies represent the largest class of protein therapeutics. Engineering efforts have focused on the following:

#### 4.1.1. Affinity Maturation

Affinity maturation is a critical process in antibody engineering that aims to enhance the binding affinity and specificity of antibodies to their target antigens [69,70]. This process typically involves iterative cycles of introducing mutations into the antibody sequence, particularly in the complementarity-determining regions (CDRs), followed by screening and selection of variants with improved binding properties. Computational approaches have become increasingly important in guiding affinity maturation efforts, with methods ranging from structure-based design to machine learning models [71,72]. Structure-guided computational design can identify promising mutations by analyzing the antibody–antigen interface and predicting energetically favorable substitutions [73]. Machine learning models, trained on experimental data and structural features, can predict the effects of mutations on binding affinity and guide the selection of candidates for experimental validation [74]. Deep learning approaches, such as those based on language models trained on antibody sequences, have shown promise in generating diverse sets of potentially affinity-enhancing mutations [75]. Experimental methods for affinity maturation include display technologies like phage and yeast display, which allow for high-throughput screening of large antibody libraries [76,77]. Advanced techniques like deep mutational scanning provide comprehensive mutational landscapes that can inform both computational and experimental approaches [54,78]. The integration of computational predictions with experimental validation in iterative cycles has emerged as a powerful strategy for efficient affinity maturation, allowing researchers to explore larger sequence spaces and identify non-obvious beneficial mutations [79,80] (Figure 3a).

#### 4.1.2. Bispecific and Multispecific Antibodies

Bispecific and multispecific antibodies represent a significant advancement in antibody engineering, enabling the creation of molecules that can simultaneously bind two or more distinct epitopes, thus facilitating novel therapeutic mechanisms [81]. These engineered antibodies can be designed to engage multiple targets on the same or different cells, offering potential advantages over traditional monoclonal antibodies in terms of efficacy and specificity [82]. Bispecific antibodies, in particular, have gained considerable attention, with over 100 candidates in clinical development and several FDA approvals since 2014. Various formats have been developed, including IgG-like structures and smaller fragments, each with distinct properties affecting pharmacokinetics, tissue penetration, and effector functions [83]. Common applications include T-cell redirection in cancer immunotherapy, where one arm binds a tumor antigen and the other engages CD3 on T cells, as exemplified by the FDA-approved blinatumomab for acute lymphoblastic leukemia [84]. Beyond oncology, bispecific and multispecific antibodies are being explored for autoimmune diseases, infectious diseases, and neurodegenerative disorders [85]. The development of these complex molecules presents unique challenges in design, production, and characterization, requiring innovative approaches in protein engineering, cell line development, and manufacturing processes [86,87]. As the field advances, computational tools, including machine learning approaches, are increasingly being employed to optimize antibody design and predict properties such as stability and aggregation propensity [88] (Figure 3a).

### 4.2. Enzyme Replacement Therapies

For genetic disorders caused by enzyme deficiencies, protein engineering has been used to:

#### 4.2.1. Enhance Enzyme Stability

Enhancing enzyme stability, particularly thermostability and resistance to proteolysis, is crucial for increasing circulatory half-life and improving the efficacy of therapeutic proteins. Thermostability can be improved through various strategies, including loop scanning and site-saturation mutagenesis to identify stabilizing mutations in flexible regions [89]. Computational approaches like structure-guided design and machine learning models can predict stabilizing mutations and guide protein engineering efforts [90]. Increasing the rigidity of enzymes, particularly in the active site region, has been shown to enhance both thermostability and proteolytic resistance [91,92]. Specific techniques to improve protease resistance include terminal modifications like N-terminal acetylation and C-terminal amidation, as well as the incorporation of non-canonical amino acids [93,94]. Cyclization of peptides and proteins can also significantly enhance stability against proteolytic degradation [95]. For larger proteins, engineering disulfide bonds or introducing covalent “staples” using non-canonical amino acids can rigidify the structure and improve thermostability [96]. Directed evolution approaches, coupled with high-throughput screening, remain powerful tools for identifying stabilizing mutations that may not be rationally predicted. By combining multiple stabilization strategies, dramatic improvements in enzyme half-life can be achieved, with some engineered variants showing 40-fold longer half-lives at elevated temperatures [92] (Figure 3b).

#### 4.2.2. Optimize Tissue Targeting

Optimizing tissue targeting by adding or modifying targeting motifs is a crucial strategy to improve enzyme delivery to affected tissues in enzyme replacement therapies. The addition of specific targeting ligands can enhance cellular uptake and tissue distribution of therapeutic enzymes [23,97]. For example, glycosylation with mannose-6-phosphate (M6P) residues enables targeting to M6P receptors that are highly expressed in many cell types, facilitating lysosomal enzyme delivery [98]. Modifying enzymes with cell-penetrating peptides like TAT or polyarginine can improve cellular uptake and tissue penetration [99]. Antibody-enzyme fusion proteins have been developed to target specific cell surface receptors and enhance tissue-specific delivery [100,101]. Nanocarrier systems functionalized with targeting ligands such as transferrin, folate, or RGD peptides can improve enzyme biodistribution to target tissues [102]. Glycoengineering approaches to modify enzyme glycosylation patterns can alter tissue tropism and pharmacokinetics [103]. Site-specific PEGylation or polymer conjugation can be used to modulate the enzyme circulation time and tissue accumulation [102]. Biomimetic strategies like erythrocyte membrane coating have shown promise for prolonging circulation and enhancing tissue-specific targeting [104]. Recent advances in protein engineering and synthetic biology are enabling the design of enzymes with intrinsic tissue-targeting properties [105,106] (Figure 3b).

### 4.3. Cytokine Engineering

Engineered cytokines offer improved therapeutic windows and targeted activity:

#### 4.3.1. Orthogonal Cytokine–Receptor Pairs

Orthogonal cytokine–receptor pairs represent an innovative approach to creating targeted therapies with reduced off-target effects by engineering both the cytokine and its receptor to interact exclusively with each other [107]. This strategy was pioneered with the development of an orthogonal IL-2/IL-2R pair (ortho2 and ortho2R) that can stimulate only engineered T cells expressing the modified receptor while remaining inert to endogenous immune cells. Similar approaches have been applied to other cytokines, such as the split, conditionally active mimetic of IL-2/IL-15 (Neo-2/15) that requires colocalization of two components for activity, enabling selective activation in specific tissue microenvironments [108]. Orthogonal engineering has also been used to create synthetic T cell states that escape canonical exhaustion, as demonstrated by T cells engineered to secrete an IL-2 variant binding the IL-2Rβγ receptor and the alarmin IL-33 [109]. These approaches can be extended to other cytokines and receptors, potentially allowing precise control of engineered therapeutic cells like CAR-T cells [110]. The development of non-natural “synthekines” that assemble non-natural receptor heterodimers represents another frontier in orthogonal cytokine engineering [111]. As the field advances, combining orthogonal cytokine–receptor pairs with engineered cellular delivery systems may offer powerful ways to autonomously target and modulate local disease environments while minimizing systemic toxicity (Figure 3c).

#### 4.3.2. Conditionally Active Cytokines

Conditionally active cytokines are engineered variants designed to be selectively activated in specific tissue microenvironments, such as tumors, to improve their therapeutic index and reduce systemic toxicity [108]. One approach involves splitting cytokines into two inactive components that require colocalization for activity, as demonstrated with a split IL-2/IL-15 mimetic that showed enhanced antitumor efficacy and reduced toxicity when the components were independently targeted [108]. Another strategy utilizes protease-activated cytokine prodrugs that leverage the dysregulated protease activity in tumors, exemplified by WTX-124, a conditionally activated IL-2 prodrug that is preferentially cleaved and activated in the tumor microenvironment [112]. pH-dependent cytokines have also been developed to exploit the acidic tumor microenvironment, with engineered variants showing selective activation under low pH conditions [113]. Some approaches focus on modifying cytokine–receptor interactions, such as engineering IL-2 variants with reduced affinity for CD25 to avoid preferential activation of regulatory T cells [114]. Targeted delivery systems, including antibody–cytokine fusions and nanoparticle formulations, can also achieve conditional activation by concentrating cytokines in desired tissues [115]. More complex designs involve multi-component systems, such as the cytokine-PACE platform, which combines split cytokines with antibody-driven chain exchange for targeted reconstitution of active cytokines on tumor cells [116]. These diverse engineering strategies aim to overcome the limitations of conventional cytokine therapies by enhancing tumor specificity and minimizing off-target effects, potentially expanding the therapeutic window for cytokine-based cancer immunotherapies [114,117,118] (Figure 3c).

#### 4.3.3. Example of AI-Driven Innovations in Biologics Design

Recent advancements in engineering techniques have significantly contributed to the design and development of next-generation biologics, leveraging computational models, machine learning, and innovative biotechnological methods. One notable example is the use of generative biology, which integrates AI and machine learning to streamline the protein drug discovery process. This approach has been shown to halve the timelines for antibody discovery and double the success rates of engineered proteins by utilizing computational models to predict viable drug candidates, which are then validated in wet labs [119]. Furthermore, cryo-electron microscopy (cryo-EM) has revolutionized structure-based drug design by enabling high-resolution structural analysis of complex biological macromolecules, such as G-protein-coupled receptors (GPCRs) and ion channels, which are critical drug targets [120]. These structural insights facilitate the design of more effective and specific biologics. Additionally, machine learning has been applied to optimize protein properties, enhancing their activity and safety while reducing the development time and costs. For instance, Amgen’s integration of machine learning models with extensive protein data has accelerated the development of multispecific drugs, which are engineered to bind multiple targets simultaneously, demonstrating the potential of computational methods to transform biologics design [119]. These examples highlight how engineering techniques are driving the innovation of biologics, making them more efficient and effective in treating various diseases.

## 5. Emerging Approaches and Future Directions

### 5.1. Intracellular Protein Delivery

Achieving efficient delivery of therapeutic proteins to intracellular targets remains a major challenge. Promising approaches include:

#### 5.1.1. Cell-Penetrating Peptides

Cell-penetrating peptides (CPPs) are short peptide sequences, typically 5–30 amino acids long, that can facilitate the cellular uptake of attached cargo proteins across biological membranes [121]. CPPs are often rich in positively charged amino acids like arginine and lysine, which interact with negatively charged cell surface molecules to promote internalization [122]. Common CPP sequences include HIV-1 Tat, penetratin, and polyarginine peptides. The mechanisms of CPP-mediated cellular entry are still debated but likely involve both direct membrane translocation and endocytic uptake [123]. A key challenge for CPP-protein delivery is escaping endosomal entrapment after internalization [124]. Some CPPs can disrupt endosomal membranes through mechanisms like the proton sponge effect or pore formation. Engineering approaches to enhance endosomal escape include incorporating pH-sensitive domains or fusogenic peptides [65]. CPPs have been successfully used to deliver a wide range of protein cargoes including enzymes, antibodies, and transcription factors both in vitro and in vivo [125]. Recent advances include the development of activatable CPPs that are triggered by tumor microenvironment conditions and cell-type specific CPPs for targeted delivery. While promising, challenges remain in optimizing the efficiency of cytosolic delivery and reducing potential toxicity of CPPs for therapeutic applications [126] (Figure 4a).

#### 5.1.2. Nanocarrier-Based Delivery

Nanocarrier-based delivery systems, including nanoparticles and liposomes, have emerged as promising approaches for protein encapsulation and targeted delivery. These nanocarriers can protect proteins from premature degradation, enhance their stability and circulation time, enable controlled release, and facilitate cellular uptake and intracellular delivery [127]. Liposomes, composed of phospholipid bilayers, can encapsulate hydrophilic proteins in their aqueous core or incorporate hydrophobic proteins in the lipid bilayer. Polymeric nanoparticles, such as those made from PLGA or chitosan, offer tunable degradation and release properties for sustained protein delivery [128]. Inorganic nanoparticles like mesoporous silica can achieve high protein loading capacity and enable stimuli-responsive release [129]. Surface modification of nanocarriers with targeting ligands or cell-penetrating peptides can enhance tissue- or cell-specific delivery [130]. Key challenges include maintaining protein stability during encapsulation, achieving efficient intracellular delivery and endosomal escape, and optimizing the nanocarrier degradation rate for controlled protein release [131]. Recent advances such as biomimetic nanoparticles and exosome-based delivery systems show promise for improving the in vivo performance of protein therapeutics. While several nanocarrier-based protein delivery systems have reached clinical trials, continued research is needed to overcome biological barriers and enhance the therapeutic efficacy of this approach (Figure 4b).

### 5.2. Stimulus-Responsive Proteins

Creating proteins that can be activated or deactivated in response to specific stimuli offers precise spatiotemporal control of therapeutic activity:

#### 5.2.1. pH-Sensitive Proteins

The acidic tumor microenvironment, characterized by extracellular pH values typically ranging from 6.5 to 6.9 compared to 7.2 to 7.4 in normal tissues, provides a unique opportunity for developing targeted cancer therapies using pH-sensitive proteins [132]. Designing such proteins involves incorporating pH-responsive elements that undergo conformational changes or altered interactions in response to acidic conditions [133]. Common strategies include introducing histidine residues, which become protonated at slightly acidic pH, into key functional regions of proteins [134]. For example, antibodies have been engineered with pH-dependent binding, allowing them to release their cargo specifically in acidic endosomes [135]. Another approach involves designing pH-sensitive protein switches using computational methods, as demonstrated by the de novo design of proteins that assemble into micron-scale fibers at neutral pH but rapidly disassemble when exposed to acidic conditions [136]. pH-responsive protein nanocarriers have also been developed, utilizing materials that change conformation or disassemble in acidic environments to release encapsulated drugs. These pH-sensitive designs can be further enhanced by combining them with other stimuli-responsive elements or targeting moieties to improve specificity and efficacy [137]. The development of conditionally active biologics (CABs) that are preferentially activated in the acidic tumor microenvironment represents another promising approach, as exemplified by protease-activated cytokine prodrugs [136]. While these strategies show great promise, challenges remain in optimizing the pH transition point, maintaining protein stability, and achieving efficient intracellular delivery for therapeutic applications (Figure 4c).

#### 5.2.2. Protease-Activated Proteins

Protease-activated proteins are engineered latent proteins that remain inactive until they are cleaved by disease-specific proteases, enabling targeted therapeutic activity. A key strategy involves designing proteins with inhibitory pro-domains that are removed by proteolytic cleavage, as demonstrated by engineered pro-forms of antibodies and cytokines that are activated in the tumor microenvironment [138,139]. Computational approaches like structure-guided design and machine learning models can predict optimal cleavage sites and pro-domain sequences to achieve desired activation profiles. Directed evolution techniques have also been applied to engineer highly specific protease recognition sequences [140]. Some designs incorporate multiple protease cleavage sites to require coincident protease activity for full activation, improving specificity [141]. Protein cages that disassemble upon protease cleavage to release encapsulated cargo offer another promising approach. Beyond oncology, protease-activated proteins are being explored for applications in infectious diseases, cardiovascular disorders, and other conditions with dysregulated protease activity [138]. Key challenges include optimizing the dynamic range between latent and active states, minimizing premature activation, and achieving efficient intracellular delivery for some applications. As the field advances, combining protease-activation with other targeting strategies may further enhance the therapeutic window of engineered proteins [139] (Figure 4d).

### 5.3. De Novo Designed Therapeutic Proteins

Advances in de novo protein design are enabling the creation of entirely novel therapeutic modalities:

#### 5.3.1. Protein Switches

Protein switches are engineered proteins that undergo programmable conformational changes in response to specific molecular cues, enabling sensing and actuation functions in synthetic biology applications. A key strategy involves designing proteins with multiple stable conformational states separated by energy barriers that can be modulated by stimuli like small molecules, pH changes, or protein–protein interactions [142]. Computational approaches have become increasingly important for switch design, with methods like structure-guided design and machine learning models enabling the prediction of conformational changes and optimization of switching behavior [143,144]. Common design principles include incorporating ligand-binding domains, using mutually exclusive folding states, and engineering allosteric coupling between distant protein regions [144,145]. Directed evolution and high-throughput screening approaches have also been successful in developing switches with desired properties. Recent advances include the de novo design of modular protein switches and sensors using approaches like deep learning. Key challenges in the field include improving the dynamic range between states, minimizing unwanted activation, and achieving rapid and reversible switching [146]. As computational and experimental methods continue to advance, protein switches are likely to play an increasingly important role in synthetic biology applications like cell-based therapeutics and biosensors [147,148].

#### 5.3.2. Artificial Enzymes

Artificial enzymes are designed catalytic proteins that aim to mimic or improve upon the functions of natural enzymes for therapeutic applications. A key strategy involves computational design of protein scaffolds with precisely positioned catalytic residues to facilitate desired reactions, as demonstrated by early work on de novo enzyme design [149]. Machine learning approaches are increasingly being used to optimize artificial enzyme designs and predict beneficial mutations. Directed evolution remains a powerful complementary method for enhancing the activity and specificity of designed enzymes [150]. For prodrug activation, artificial enzymes have been engineered to catalyze bond-forming reactions that construct a drug’s pharmacophore, enabling highly selective activation [151]. Incorporation of non-canonical amino acids or metal cofactors can introduce novel catalytic functionalities not found in natural enzymes [152,153]. Artificial metalloenzymes combining protein scaffolds with synthetic metal catalysts show promise for expanding the repertoire of biocompatible reactions [154]. Computational design of enzyme switches that are selectively activated by disease-specific triggers offers another avenue for targeted therapeutics [155]. While significant progress has been made, challenges remain in achieving the catalytic efficiency and substrate specificity of natural enzymes [156,157]. Key hurdles include improving in vivo half-life, enhancing targeted action, and controlling immune responses to enzyme therapeutics [154]. Continued advances in computational methods, high-throughput screening, and our understanding of enzyme mechanisms will be key to realizing the full potential of artificial enzymes as therapeutics [148].

## 6. The Synergy of High-Throughput Screening and Structural Studies

The integration of computational methods with high-throughput screening (HTS) and structural studies has revolutionized drug discovery, providing a robust and dynamic interface that enhances the identification and optimization of therapeutic candidates. HTS remains a cornerstone in early drug discovery, enabling the rapid screening of vast compound libraries to identify active molecules. Recent advancements in HTS, particularly in mass spectrometry and automation, have expanded its capabilities, making it more cost-effective and physiologically relevant [158]. This high-throughput approach, when coupled with structural studies such as cryo-electron microscopy and X-ray crystallography, provides detailed insights into the binding interactions and conformational dynamics of target proteins, which are critical for the rational design of drugs [159,160]. Computational tools, including AI-driven platforms like AlphaFold and Rosetta, complement these experimental techniques by predicting protein structures and interactions, thereby guiding the selection and optimization of hits identified through HTS [161]. The synergy between computational predictions and experimental validation is particularly crucial for understanding allosteric binding sites, which are often difficult to predict computationally but can be uncovered through functional screening [162]. This integrated approach not only accelerates the drug discovery process but also enhances the accuracy and efficacy of therapeutic candidates, demonstrating the indispensable role of combining computational and wet lab methodologies in modern drug discovery.

## 7. Challenges and Future Outlook

While protein engineering has made tremendous strides, several challenges remain. Improving our ability to predict protein behavior in complex physiological environments remains a major challenge for engineered protein therapeutics [163]. Developing scalable manufacturing processes for increasingly complex engineered proteins will be crucial as more sophisticated designs enter clinical development [164]. Enhancing immunogenicity prediction and mitigation strategies is critical to reduce adverse immune responses that can limit efficacy and safety [165]. Improving targeted delivery methods to specific tissues and cell types could dramatically enhance therapeutic index and enable new applications. Designing protein therapeutics that synergize effectively with other treatment modalities like small molecules or cell therapies offers exciting potential for combination approaches. Addressing these challenges will require continued advances in computational modeling, high-throughput screening methods, protein engineering techniques, drug delivery technologies, and systems biology approaches to understand complex in vivo environments.

In the future, the combination of sophisticated computational tools, high-throughput experimental techniques, and a better knowledge of how protein structure and function are related will continue to lead to advancements in therapeutic protein engineering. An effective and dynamic connection between computational and wet lab research is essential for determining function. Ultra-high-throughput screening, such as in this case, continues to be a cost-effective and impartial technique for discovering initial stages for engineering. By integrating structural investigations, computational techniques can optimize these initial points to improve the design and effectiveness of protein treatments. Upcoming technologies, such mRNA therapies and in vivo protein evolution, are ready to enhance the potential for developing and delivering modified proteins. As these methods develop, we should expect a new wave of customized protein treatments that have enhanced effectiveness, safety, and usefulness for a variety of disorders.

## 8. Conclusions

Protein design and engineering have become indispensable tools in the development of advanced biotherapeutics. By harnessing computational and experimental approaches, researchers are creating protein drugs with enhanced stability, efficacy, and safety profiles. As our ability to manipulate protein structure and function continues to improve, we can anticipate a future where highly customized protein therapeutics provide precise and powerful treatments for a wide range of human diseases.

## Figures and Tables

**Figure 1 biomolecules-14-01073-f001:**
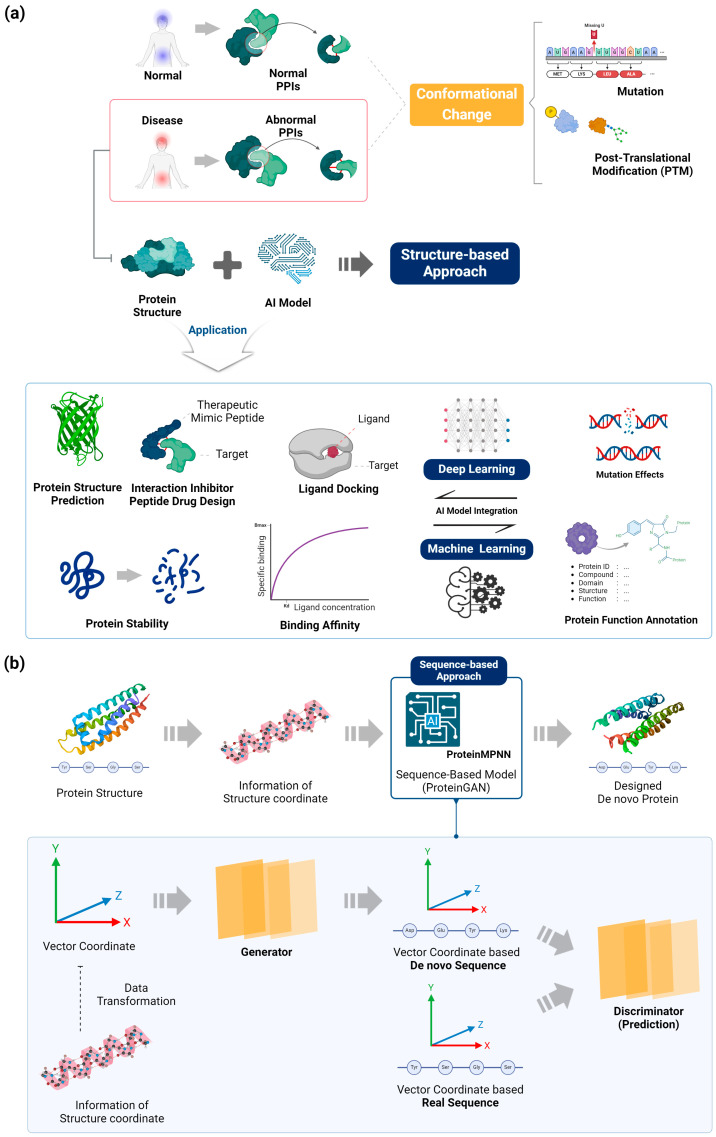
(**a**) Overview and applications of structural proteome-based integrated artificial intelligence. Structure-based computational design has advanced dramatically through the integration of existing machine learning and deep learning algorithms. It can now be used in various fields, including protein structure and enzyme design, ligand docking, structure prediction of biological macromolecules and complexes, and protein function annotation. The integration of these algorithms and the advancement of structure-based computational techniques contribute to the optimization and advancement of structural biology for therapeutic protein engineering applications. (**b**) The sequence-based approach is emerging as a powerful tool for protein design, instrumental in the development of therapeutic drugs. ProteinMPNN, a representative sequence-based graph neural network model, utilizes a generative algorithm and has demonstrated excellent performance in redesigning proteins by leveraging information about the protein backbone. Consequently, using sequence-based models, researchers are able to design de novo proteins with enhanced stability and binding affinity.

**Figure 2 biomolecules-14-01073-f002:**
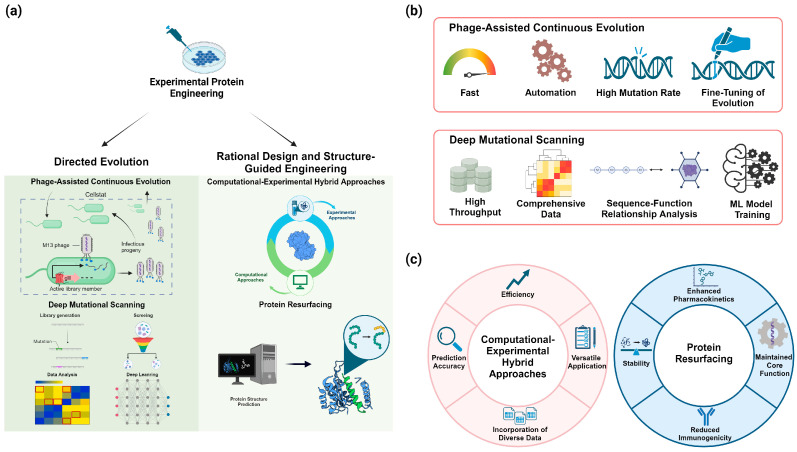
Experimental protein engineering has achieved significant advancements through directed evolution as well as rational design and structure-guided engineering. (**a**) A schematic diagram explaining Phage-Assisted Continuous Evolution and Deep Mutational Scanning within Directed Evolution, and Computational–Experimental Hybrid Approaches and Protein Resurfacing within Rational Design and Structure-Guided Engineering. (**b**) A comparison highlighting the advantages of Phage-Assisted Continuous Evolution and Deep Mutational Scanning compared to traditional methods. (**c**) A detailed description of the advantages of Computational–Experimental Hybrid Approaches and Protein Resurfacing compared to traditional methods.

**Figure 3 biomolecules-14-01073-f003:**
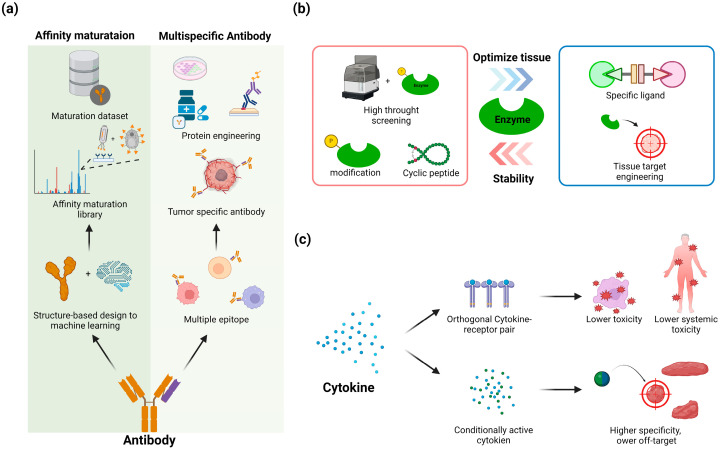
An overview and applications of therapeutic protein engineering using advanced biological molecules. (**a**) The workflows of protein engineering using antibodies, which can be applied through computational and experimental methods. (**b**) An overview of the development direction of protein engineering for enzyme stability and tissue optimization. (**c**) The workflow of developing lower toxicity and off-target effects using cytokine engineering.

**Figure 4 biomolecules-14-01073-f004:**
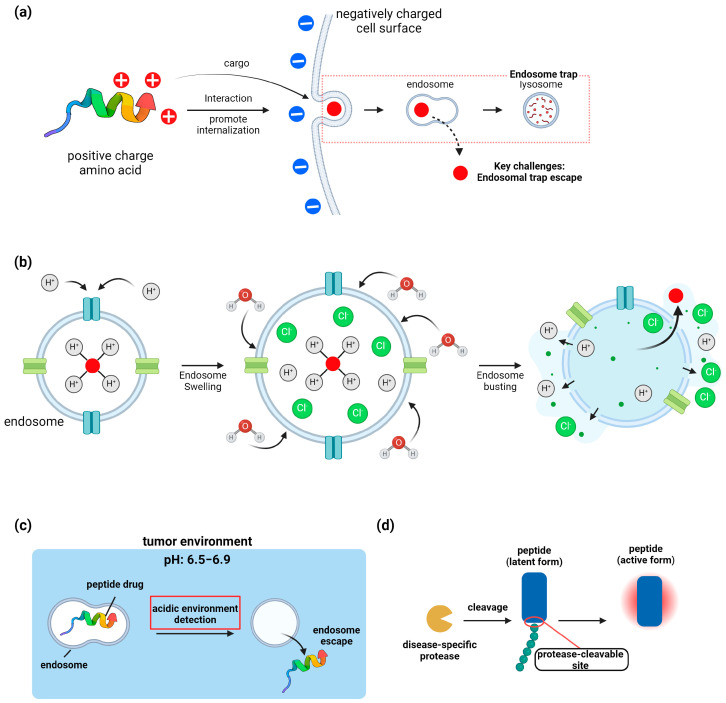
Mechanisms of cell-penetrating peptide (CPP) delivery and endosomal escape. (**a**) Mechanism of cellular membrane penetration by a cell-penetrating peptide (CPP) containing positively charged amino acids, and subsequent evasion of the endosome trap. (**b**) Schematic of the proton sponge mechanism, one of the mechanisms for the escape of peptides from endosomal membranes. The diagram illustrates the proton sponge mechanism, where the peptide captures protons leverage osmotic pressure to facilitate the influx of water molecules into the endosome. This influx induces endosomal swelling, culminating in the rupture of the endosome and the subsequent release of the peptide. (**c**) Escape of peptide drugs from the endosome upon detection of the acidic environment of a tumor. (**d**) Diagram depicting the activation of a latent form peptide by a disease-specific protease via cleavage.

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
