# Peer review of "Integrating Computational Design and Experimental Approaches for Next-Generation Biologics"

_biomolecules, 2024, doi:10.3390/biom14091073_

Round 1

Reviewer 1 Report

Comments and Suggestions for Authors

The manuscript provides a comprehensive review of protein engineering methods and advancements, introducing each technique and concept effectively. Its clarity and structure make it accessible to researchers across various fields. However, the review could be enhanced by addressing the following points:

1.     From my perspective, the manuscript is divided into two main parts: engineering techniques and next-generation biologics. Currently, the connection between these two segments is somewhat weak. To strengthen this link, consider elaborating on how specific engineering techniques contribute directly to the design of next-generation biologics by adding detailed examples of successful applications within Chapter 4 and 5.

2.     Besides the examples, I also would like to see more comparisons between methods in each segment, instead of only listing and briefly introducing each method. For example, a thorough comparison between sequence-based and structure-based design, outlining their differences, advantages, and disadvantages, would be beneficial. Similarly, a side-by-side analysis of Rosetta and AlphaFold would enhance the depth of the review.

3.     Ensure that all figures are referenced in the text.

4.     The narrative flow in the introduction section is somewhat disrupted between the 3rd, 4th, and 5th paragraphs. To ensure coherence, it would be beneficial to start the fifth paragraph with a sentence that not only highlights exciting techniques but also connects these to the development of next-generation biologics, such as bispecific and multispecific antibodies. This addition would help bridge the concepts introduced earlier and provide a smooth transition into antibody engineering.

Author Response

Responses to Reviewers #1 (biomolecules-3146379)

[1] Comments for the Author:

From my perspective, the manuscript is divided into two main parts: engineering techniques and next-generation biologics. Currently, the connection between these two segments is somewhat weak. To strengthen this link, consider elaborating on how specific engineering techniques contribute directly to the design of next-generation biologics by adding detailed examples of successful applications within Chapter 4 and 5.

Author’s Response:

Thank you for your insightful feedback. In response to the reviewers' suggestions, we added an additional paragraph between Chapters 4 and 5 to describe how specific engineering technologies directly contribute to the design of next-generation biologics. Specifically, in our manuscript revision, on page 12, lines 482-501, we state:

“Recent advancements in engineering techniques have significantly contributed to the design and development of next-generation biologics, leveraging computational models, machine learning, and innovative biotechnological methods. One notable example is the use of generative biology, which integrates AI and machine learning to streamline the protein drug discovery process. This approach has been shown to halve the timelines for antibody discovery and double the success rates of engineered proteins by utilizing computational models to predict viable drug candidates, which are then validated in wet labs. Furthermore, cryo-electron microscopy (cryo-EM) has revolutionized structure-based drug design by enabling high-resolution structural analysis of complex biological macromolecules, such as G protein-coupled receptors (GPCRs) and ion channels, which are critical drug targets. These structural insights facilitate the design of more effective and specific biologics. Additionally, machine learning has been applied to optimize protein properties, enhancing their activity and safety while reducing development time and costs. For instance, Amgen's integration of machine learning models with extensive protein data has accelerated the development of multispecific drugs, which are engineered to bind multiple targets simultaneously, demonstrating the potential of computational methods to transform biologics design. These examples highlight how engineering techniques are driving the innovation of biologics, making them more efficient and effective in treating various diseases.”

[2] Comments for the Author:

Besides the examples, I also would like to see more comparisons between methods in each segment, instead of only listing and briefly introducing each method. For example, a thorough comparison between sequence-based and structure-based design, outlining their differences, advantages, and disadvantages, would be beneficial. Similarly, a side-by-side analysis of Rosetta and AlphaFold would enhance the depth of the review.

Author’s Response:

We appreciate the reviewers' feedback. In response to their suggestions, we added an additional paragraph to describe a comparative analysis of sequence-based versus structure-based drug design. In our manuscript revision, on page 6, lines 226-245, we state:

“Recent advancements in drug discovery have highlighted the distinct methodologies and applications of sequence-based and structure-based design approaches, each with its unique strengths and limitations. Sequence-based drug design, such as the sequence-to-drug concept validated by TransformerCPI2.0, leverages protein sequence data to predict drug candidates through end-to-end differentiable learning models. This method is particularly advantageous for targets lacking high-resolution 3D structures, as demonstrated by its success in identifying novel modulators for proteins like SPOP and RNF130, which are challenging to model structurally. On the other hand, structure-based drug design (SBDD) relies on the detailed 3D structures of target proteins to identify binding sites and optimize drug interactions. Recent advances in computational algorithms and techniques like cryo-electron microscopy have significantly enhanced the precision of SBDD, enabling the design of highly specific and effective biologics. However, SBDD is often limited by the availability and accuracy of protein structures and the complexity of modeling dynamic binding sites. While sequence-based methods offer a more direct and potentially faster route to drug discovery, structure-based approaches provide deeper insights into molecular interactions, crucial for optimizing drug efficacy and safety. The integration of both methods, leveraging the strengths of each, holds promise for advancing the development of next-generation therapeutics, offering a comprehensive toolkit for tackling diverse drug discovery challenges.”

We also performed a side-by-side analysis of the differences between Rosetta and AlphaFold and revised the manuscript accordingly. On pages 4-5, lines 139-162, we state:

“AlphaFold and Rosetta are two leading tools in the field of protein structure prediction, each with distinct methodologies and applications. AlphaFold, developed by DeepMind, employs a deep learning approach that leverages sequence coevolution data to predict the distance and angle between amino acids, achieving remarkable accuracy in protein folding predictions. This method has demonstrated exceptional performance in the Critical Assessment of Structure Prediction (CASP) competitions, with AlphaFold2 achieving a median Global Distance Test (GDT) score of 92.4, close to experimental accuracy. In contrast, Rosetta, created by the Baker lab, uses a combination of physics-based and knowledge-based methods, employing Monte Carlo algorithms to sample protein conformations and score them based on their probability. Rosetta is known for its flexibility and has been extensively used for protein design, docking, and related tasks. While AlphaFold excels in predicting monomeric protein structures with high accuracy, Rosetta offers robust performance in modeling protein complexes and quaternary structures, especially when supplemented with experimental data such as covalent labeling. However, AlphaFold's reliance on neural networks can sometimes lead to inaccuracies in loop regions and dynamic binding sites, where Rosetta's physics-based approach may provide more reliable results. Despite AlphaFold's groundbreaking accuracy, it occasionally struggles with predicting the structural impact of point mutations and conformational ensembles, areas where Rosetta's energy-based scoring and conformational sampling can offer more detailed insights. Overall, while AlphaFold represents a significant advancement in AI-driven protein structure prediction, Rosetta's comprehensive toolkit and integration with experimental data make it a valuable complement, particularly for complex and dynamic protein systems.”

[3] Comments for the Author:

Ensure that all figures are referenced in the text.

Author’s Response:

Thank you for the advice. We have referenced all figures in the text as per the reviewers' comments.

[4] Comments for the Author:

The narrative flow in the introduction section is somewhat disrupted between the 3rd, 4th, and 5th paragraphs. To ensure coherence, it would be beneficial to start the fifth paragraph with a sentence that not only highlights exciting techniques but also connects these to the development of next-generation biologics, such as bispecific and multispecific antibodies. This addition would help bridge the concepts introduced earlier and provide a smooth transition into antibody engineering.

Author’s Response:

Thank you for your valuable feedback. We revised the fifth paragraph of the introduction to begin with a sentence that connects it to the development of next-generation biologics. In our manuscript revision, on page 2, lines 64-68, we state:

“Building upon these advancements, the development of bispecific and multi-specific antibodies exemplifies the potential of therapeutic protein engineering to create next-generation biologics capable of addressing complex disease mechanisms. These engineered proteins can simultaneously bind to multiple targets, offering innovative solutions in areas such as cancer immunotherapy and other multifactorial diseases.”

Reviewer 2 Report

Comments and Suggestions for Authors

A well researched and useful review, which captures key developments in the field.

One suggestion: Perhaps make more of the importance of a strong and dynamic interface between computational and wet lab experimentation, especially in relation to determining function.  Ultra high throughput screening still represents a cost effective and unbiased way to find starting points for engineering, and coupled with structural studies enables computational approaches to be successful.  This is mentioned in lines 72 and 580 and in Figure 3b, but its importance is under-represented in my view.  The binding of functional hits from screening can surprise on the upside for example, binding allosterically and very difficult to predict computationally.

Author Response

Responses to Reviewers #2 (biomolecules-3146379)

[1] Comments for the Author:

One suggestion: Perhaps make more of the importance of a strong and dynamic interface between computational and wet lab experimentation, especially in relation to determining function. Ultra high throughput screening still represents a cost effective and unbiased way to find starting points for engineering, and coupled with structural studies enables computational approaches to be successful.  This is mentioned in lines 72 and 580 and in Figure 3b, but its importance is under-represented in my view.  The binding of functional hits from screening can surprise on the upside for example, binding allosterically and very difficult to predict computationally.

Author’s Response:

Thank you for your insightful feedback. In response to the reviewers' suggestions, we wrote an additional paragraph to describe the importance of a strong and dynamic interface between computational and wet lab experimentation. In our manuscript revision, on page 16, lines 643-662, we state:

“The integration of computational methods with high-throughput screening (HTS) and structural studies has revolutionized drug discovery, providing a robust and dynamic interface that enhances the identification and optimization of therapeutic candidates. HTS remains a cornerstone in early drug discovery, enabling the rapid screening of vast compound libraries to identify active molecules. Recent advancements in HTS, particularly in mass spectrometry and automation, have expanded its capabilities, making it more cost-effective and physiologically relevant. This high-throughput approach, when coupled with structural studies such as cryo-electron microscopy and X-ray crystallography, provides detailed insights into the binding interactions and conformational dynamics of target proteins, which are critical for the rational design of drugs. Computational tools, including AI-driven platforms like AlphaFold and Rosetta, complement these experimental techniques by predicting protein structures and interactions, thereby guiding the selection and optimization of hits identified through HTS. The synergy between computational predictions and experimental validation is particularly crucial for understanding allosteric binding sites, which are often difficult to predict computationally but can be uncovered through functional screening. This integrated approach not only accelerates the drug discovery process but also enhances the accuracy and efficacy of therapeutic candidates, demonstrating the indispensable role of combining computational and wet lab methodologies in modern drug discovery.”

Round 2

Reviewer 1 Report

Comments and Suggestions for Authors

The new version addresses most of my concerns, and I agree to proceed with publication.